# Reconstructive Options for Pressure Ulcers in Pediatric Patients

**DOI:** 10.3390/children11060691

**Published:** 2024-06-05

**Authors:** Dominika Krakowczyk, Jakub Opyrchał, Tomasz Koszutski, Krzysztof Dowgierd, Łukasz Krakowczyk

**Affiliations:** 1Pediatric Surgery and Urological Department, Upper Silesian Child Health Center in Katowice, Silesian University of Medicine, 40-052 Katowice, Poland; tkoszutski@sum.edu.pl (T.K.); lukasz.krakowczyk@gliwice.nio.gov.pl (Ł.K.); 2Department of Oncological and Reconstructive Surgery, Maria Sklodowska Curie Memorial National Cancer Center, 44-100 Gliwice, Poland; jakub.opyrchal@io.gliwice.pl; 3Head and Neck Surgery Clinic for Children and Young Adults, Department of Clinical Pediatrics, University of Warmia and Mazury, 10-082 Olsztyn, Poland; krzysztof.dowgierd@uwm.edu.pl

**Keywords:** pressure ulcers, reconstructive surgery, pediatric patients

## Abstract

Background: Pressure ulcers pose significant challenges in terms of treatment, often exhibiting a low success rate and a propensity for recurrence. Children with neurological impairments such as myelomeningocele and those with spinal injuries are particularly vulnerable to developing pressure ulcers. Despite advancements, achieving successful reconstruction remains a formidable task. Common sites prone to pressure ulcer formation include the sacral and ischial regions, as well as areas over bony prominences. Additionally, pressure ulcers attributable to medical devices facilitating ambulation are observed. While many pressure sores resolve spontaneously, conservative management may prove ineffective for some, especially in cases of stage 3 and 4 ulcers, necessitating surgical intervention. Various surgical techniques are employed for the treatment of decubitus ulcers, yet there exists no universally accepted gold standard for their management. This paper presents our institutional experience in this domain, highlighting differences in surgical approaches, treatment outcomes, complication rates, and long-term follow-up. Methods: This study involved a retrospective analysis of medical records from 11 children, ranging in age from 10 to 17 years, who presented with extensive pressure ulcers that were unresponsive to conservative treatment measures. Data collection spanned from February 2017 to June 2022. The pressure ulcers affected various anatomical regions, including the ischial area (5/11 patients), sacral region (3/11 patients), lower limb (1/11 patients), elbow (1/11 patients), and perineal area (1/11 patients). Surgical intervention was the chosen approach for all cases, employing techniques such as reconstructive surgery utilizing perforator, pediculated flaps, and locoregional flaps. Results: Eleven patients with sore ulcers (stage 3 and 4) were treated surgically. We present our experience of using surgical methods, including pedicled anterolateral flaps, pedicled gracilis musculocutaneous flaps, propeller flaps and locoregional flaps. In some cases, surgery was performed after 60 days of hospitalization or ten years after ulcer occurrence. We reviewed the length of hospital stay, surgical management and patient satisfaction. Patients were followed up to 5 years post-surgery. All flaps survived except for one flap where partial necrosis was observed. The recurrence rate was 9.01% (1/11). One patient underwent another surgery. The general outcome was satisfactory. Conclusions: Conclusions: Our findings underscore the efficacy of flap reconstruction surgical techniques in the management of pressure ulcers among pediatric patients. Based on our experience and the outcomes observed, we advocate for considering reconstructive surgery as a viable therapeutic option early in the treatment course, particularly for stage 3 and 4 ulcers. This approach not only addresses the immediate needs of patients but also holds promise for long-term wound healing and prevention of recurrence.

## 1. Introduction

The occurrence of pressure ulcers results from tissue compression and subsequent tissue ischemia that can lead to necrosis of the skin and the underlying tissues, most frequently occurring over a bony prominence. Pressure ulcers are often caused by staying too long in one position. Their occurrence can be also related to the pressure induced by medical devices, such as wheelchairs, whose aim is to facilitate ambulation. The prevalence of ulcers seems to be higher in adults and geriatric population. However, it can also affect children. The incidence rate in the adult population is estimated at about 8.8–85% [1] (20% in paraplegic patients and 86% in intensive care patients). The incidence rate in children ranges from 1.4% to 26%, in hospital 35%. In some cases, it reaches even 50% in pediatric intensive care units [2,3,4]. Since there is no good protocol or algorithm for children, some strategies that are routinely implemented in adult population must also be used in pediatric patients.

Wheelchair-bound patients and bed-bound children are susceptible to the development of pressure ulcers due to an inappropriate mattress, malnutrition, inadequate hydration and neurological impairment [5]. According to the European Pressure Ulcer Advisory Panel (EPUAP) classification system, there are four stages of pressure ulcers. Stage 4, which is the most severe, involves the skin, fat tissue, muscle, and bone structures. Such decubitus ulcers are resistant to conservative treatment. Surgical management of pressure sores not only helps in the prevention of progressive osteomyelitis, but also reduces the recurrence rates and improves the quality of life [6].

Most stage 1 and 2 pressure ulcers resolve spontaneously. However, treatment of stage 3 and 4 ulcers is still challenging. It generates high costs, prolongs hospitalization, and is characterized by recurrence. Additionally, some ulcers are associated with a 2-fold increase in mortality [6].

All kinds of pressure ulcers required good treatment including debridement, which involves removal of dead, damaged and often infected tissue to improve the healing potential of the remaining healthy tissue. We performed it using surgical, mechanical, and chemical methods. Only debridement and dressing changes comprise what we call conservative treatment in this paper.

Pressure ulcers that do not respond to conservative treatment may undergo surgical flap reconstruction. Each pressure sore should have copious irrigation and debridement of any foreign material and devitalized tissue. If nonviable or infected, bony prominences responsible for pressure injuries should be removed, or reduced in volume and appropriately modeled [5].

According to the reconstructive ladder, different techniques are used to restore the damaged tissue. They include primary wound closure, skin grafting, skin/muscle or musculocutaneous flaps, pedicled flaps, and microvascular free flaps.

A pedicled flap is a portion of tissue that maintains its vascular supply by preservation of the vessels nourishing the intact flap. When a pedicled flap is harvested from the area surrounding the defect to be covered, it is called a local flap. If it is harvested at some distance from the defect that it is reached by means of a longer pedicle, it is known as a regional flap or perforator-based propeller flap [5,7]. In turn, musculocutaneous flaps can provide adequate bulk to obliterate “dead space” after debridement of nonviable tissue and close the skin without tension [8].

This study shows the clinical experience of our team related to the treatment of pressure sores using various reconstructive procedures. Despite the considerable experience of our department in performing free flap microsurgical procedures, we presented solutions that do not require microanastomoses to make them more accessible for departments that do not possess a microscope or do not have a team experienced in microsurgery.

## 2. Materials and Methods

A retrospective case series analysis of 11 consecutive patients with the mean age of 14.4 years (range 9–17 years) was performed. The medical records of patients were collected between 2017 and 2022. Seven subjects were diagnosed with meningomyelocele. Two of them were paraplegic due to spinal injury, one patient presented with limb amputation, and one subject was diagnosed with tetraplegia due to cerebral palsy. All patients were bed-bound and underwent surgical reconstruction of stage 4 pressure sores using pedicled anterolateral thigh flaps (pALTF), pedicled musculocutaneous gracilis flaps or other locoregional flaps.

In most cases, regional flaps were sufficient to obtain safe and durable soft tissue coverage. It is worth noting that a microsurgical flap is not routinely performed in our center as the first line treatment. It is used when other methods fail.

The mean postoperative follow-up in all patients was 2 years (range: 5–60 months). All patients were treated before surgical intervention in hospital, and the mean length of hospital stay was 60 days (range: 30–180 days). The wounds were treated with conservative methods, including surgical debridement and different wound dressings. The negative pressure wound therapy (NPWT) was applied in most cases with nutritional counselling and education about pressure relief.

Pressure ulcers were debrided using a surgical knife or hydrosurgical debridement and the subsequent NPWT was used. Its effect on promoting wound healing is widely accepted by clinicians.

Exudative fluid was drained. Blood flow was increased in the wound with the formation of granulation tissue in the wound bed, which significantly shortened the time of wound preparation for the reconstructive procedure. Wound swabs were sent for culture and sensitivity before surgery. In most cases, culture and sensitivity were negative. All patients underwent debridement, including the protruding bone. Bony prominences responsible for the pressure injuries were removed if nonviable or contaminated, or eventually diminished for smaller shape. NPWT was used to prepare the wound bed, which reduced bacterial contamination, removed secretions, promoted granulation, and reduced the size of the defect.

Based on BMI-for-age percentiles, four patients were underweighted, six subjects were of normal weight, and two patients were either overweight or obese. Nutritional laboratory results were not available.

The severe pressure ulcers or refractory for conservative treatment were treated with flap coverage.

A pedicled flap (skin, adipose tissue, muscle, and sometimes fascia) is moved from donor site, its blood vessels are transferred, and the tissue is tunneled underneath the flap.

Pedicled anterolateral thigh flaps were used for the reconstruction after identification of the vessels using hand-held Doppler. The flap consisted of the skin, subcutaneous tissue, part of the vastus lateralis muscle, and the fascia. The skin paddle should be designed more distally to increase the pedicle length and arc of rotation. Some soft-tissue portions of tissue around the pedicle can be preserved to prevent kinking and spasm of the vasculature. The flap was transferred to the defect through a subcutaneous tunnel. An adequate subcutaneous tissue tunnel needs to be created to accommodate the pedicle without compression. Part of the bone was resected when infection occurred and the necrosis of the closest bone tissue was reported (as confirmed by MRI). Other infected tissues, e.g., surrounding muscles or bursae, were also removed. The tissue defect was closed in layers to eliminate the dead space and reduce tension caused by insertion of flaps.

The drains were left in the wound and in the donor site. The drain was removed depending on the amount of fluid, on average, after 3 days in the donor site (range: 2–5 days), and, on average, after 12 days in the recipient site (range: 7–24 days). The donor site was closed primarily in all cases.

Harvesting of the gracilis flap is different. The preoperative design of the flap is based on a line joining the pubic tubercle with the medial tibial condyle (the posterior borer of the flap is the adductor longus, which is easily recognized during adduction of the leg). The length of the flap is usually not more than 15 cm, and width is around 7 cm. After dissection of the whole muscle, the pedicle was transferred in the large subcutaneous tunnel. Again, the flap donor site was primarily closed.

The viability of the flap was assessed based on an evaluation of the color of the skin island, capillary return, and hand-held Doppler examination of the supplying vessels. Postoperative complications included dehiscence, hematomas, seromas, infections, and partial flap necrosis. Recurrence was found only in one female patient with an ischial ulcer.

We ask the following research question: When should we perform surgical flap cover, is it always liable, and is it the definitive solution?

## 3. Results

Between 2017 and 2022, patients with pressure ulcers in ischial (Figure 1), sacrococcygeal (Figure 2), lower leg (Figure 3) and elbow regions (Figure 4) underwent resection combined with pALTF (Figure 5), or pedunculated musculocutaneous gracilis flaps (Figure 6) and locoregional flaps (Figure 7) in the Department of Pediatric Surgery and Urology. Table 1 shows the patient characteristics and other details of the treatment.

All ulcers were surgically debrided. Necrotic tissues were completely excised. Reconstruction was another stage of treatment. Patients were advised not to bear weight on the operated region for 3–4 weeks postoperatively and gradually increase weight bearing on the reconstructed area.

No complications were reported at the donor site (Figure 8). However, we observed one partial necrosis of the flap (Figure 9) and partial wound dehiscence in three patients (it was treated conservatively and with hyperbaric therapy (Figure 10)). All flaps survived (100%). The recurrence rate was 9.01% (1/11). Another surgery was performed in one female patient due to a recurrent pressure ulcer probably caused by seroma (Figure 11). This recurrence was reported during the 5-year follow-up and the surgical procedure was performed after 11 months. Therefore, the chosen intervention was a pedicled anterolateral thigh (pALT) fasciocutaneous flap reconstruction for the ischial ulcer via a subcutaneous route.

The answer to the research question is as follows: we should not wait with the flap surgical cover, it is always worth trying.

Figure 12 shows an algorithm of a pediatric patient with pressure ulcer:

## 4. Discussion

Apart from surgical debridement and subsequent granulation, pressure ulcers refractory to nonoperative treatment may undergo flap reconstruction.

We treated stage 3 and 4 pressure ulcers, which involved the skin, fat tissue, muscle, and bone structures. Pressure sores, particularly stage IV ulcers, pose a serious challenge to surgeons. They are very difficult to treat and are associated with costs for patients and medical centers. Pressure ulcers are characterized by a tendency to recur and are associated with a more than 2-fold increase in mortality due to indirect causes (e.g., systemic infections). Surgical management of pressure sores helps in the prevention of progressive osteomyelitis and leads to an improvement in quality of life. Patients with stage 3 and 4 ulcers are at higher risk of death [5].

Even though reconstructive surgery is commonly used in the treatment of pressure ulcers there are no randomized studies on the advantages and risks related to surgical treatment or the selection of an appropriate technique [7,9,10].

The benefits of vascularized soft tissue transfer are well known for various complex chronic wounds in adults [11]. There are only a few publications about them in the pediatric population and the success rates are as low as 60%. Furthermore, the long-term rates of recurrent pressure ulcers can be high and range from 20% to even 80% [2,12,13].

In contrast to the recurrence rates reported for the surgical repair of pressure sores in the adult population, the recurrence rate in the pediatric population is significantly lower (5%). This demonstrates that surgical reconstruction of pressure sores in pediatric patients can be successful and provide long-term skin integrity [2,14].

A review of the literature shows 12 methods that can be applied for the reconstruction or treatment of ischial pressure sores. They are as follows: inferior gluteus maximus island flap [15], hamstring myocutaneous flap [16], gluteal thigh fasciocutaneous flap [17], gracilis myocutaneous island/gracilis muscle flap [18,19,20], adipofascial turnover and fasciocutaneous flap [21], tensor fascia lata flap [22], inferior gluteal artery perforator (IGAP) flap [23], lateral thigh V-Y fasciocutaneous flap, anterior thigh flap [24], rectus abdominis myocutaneous flap [25], and adductor muscle perforator flap [26].

Most papers have reported the advantages of the superior gluteal artery perforator (SGAP) flap because it can be a sensate flap. It may be useful for the reconstruction of sore ulcers in the sacral region. However, it is unlikely to be used in the ischial region because of affected muscles of the gluteal region. Additionally, it is not suitable for patients with paraplegia who can easily flex their hips and create tension on the ischial wounds because the wound will dehisce [18,27,28].

A pALTF has a lot of advantages, including constant blood supply and good bulk, and it can be harvested with the skin island, fascia, part of the muscle, and subcutaneous tissue. However, it is a non-sensate flap and sensory loss is one of the most important risk factors for pressure ulcer development and recurrence [2,27].

Long-term surgical outcomes of pressure ulcer management can be diminished by noncompliant patients. According to some surgeons, recurrence is not secondary to the surgical technique, but results from poor compliance at home or the lack of proper wound care [18,29]. Patients and their relatives must be educated about pressure relief and adequate skin care, including hydration. Ischial pressure sore patients must learn pressure-release maneuvers that should be performed periodically while sitting. Additionally, it is important to perform a daily examination of the trochanteric, ischial, sacral, and heel areas. Early recognition and treatment are mandatory to prevent progression and accelerate ulcer healing [29]. Despite excellent surgical treatment, the recurrence rate still remains at the level of 5% to 42% [1,2,4,24].

Another question is the use of musculocutaneous flaps to fill the dead space. Some authors have shown that the transferred muscle becomes atrophic, loses its dynamic function and no longer functions as a cushion to absorb pressure over time [8].

In their study on 94 patients, Thiessen et al. found no correlation between the type of the flap and the recurrence rates even if the muscle became atrophic [8]. According to the literature, sensate flaps are recommended for surgical treatment of pressure ulcers [27]. In another study, long-term outcomes of surgical reconstruction of pediatric pressure ulcers were described. In the study, 19 paraplegic patients (aged 9–16 years) with ulcers localized in sacral (n = 7), ischial (n = 9), trochanteric (n = 3), and iliac crest (n = 1) regions were treated surgically with myocutaneous flaps. The overall pressure sore recurrence rate after treatment was 5%. The long-term follow-up over the 5-year period showed that the treatment was successful and provided long-term skin integrity [4].

## 5. Conclusions

Our surgical treatment of pressure ulcers seems to be the method of choice in the management of stage 3 and 4 pressure ulcers. The ease of obtaining the flap and its good vascularization ensure the correct reconstruction of the anatomical structures with a low possibility of complications.

Surgical techniques using flap reconstruction can be beneficial in the management of pressure ulcers in pediatric population. Our experience and the results suggest that reconstructive surgery should be a therapeutic option at the beginning of treatment of stage 3/4 ulcers.

## Figures and Tables

**Figure 1 children-11-00691-f001:**
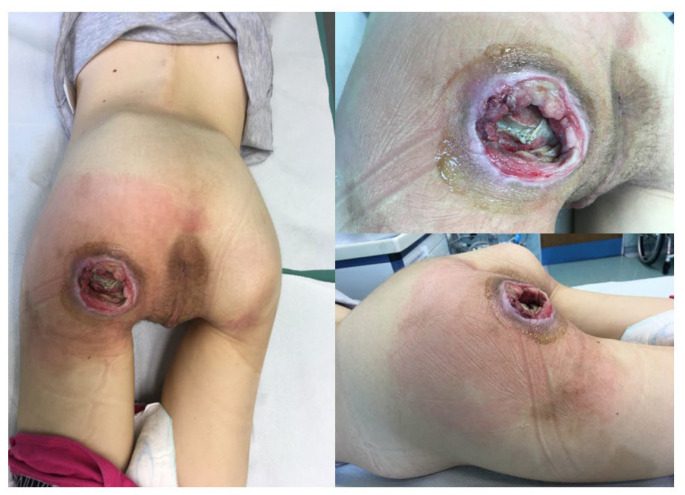
The patient with pressure ulcers in ischial region.

**Figure 2 children-11-00691-f002:**
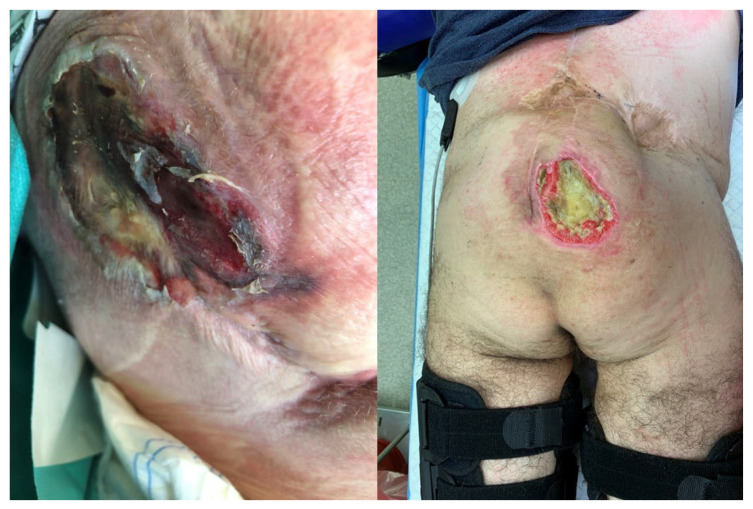
The patient with pressure ulcer in sacrococcygeal region.

**Figure 3 children-11-00691-f003:**
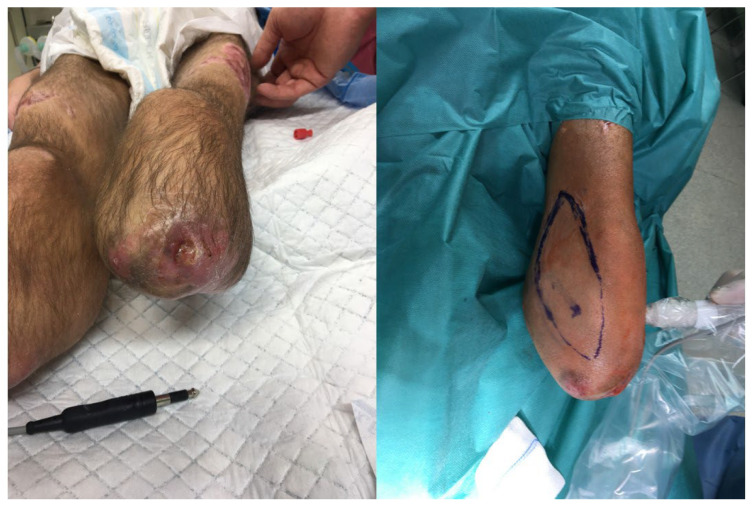
The patient with lower leg PU after legs amputation.

**Figure 4 children-11-00691-f004:**
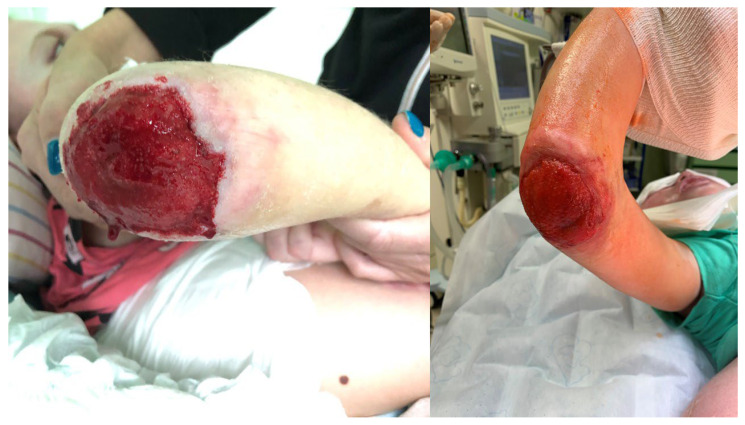
The patient with elbow region sore ulcer.

**Figure 5 children-11-00691-f005:**
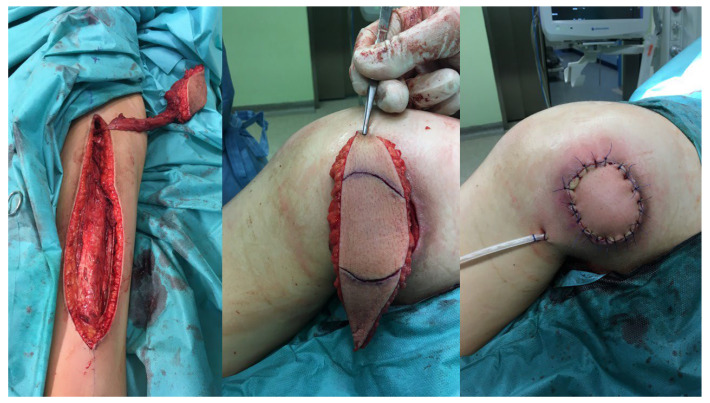
The patient with treatment the PU with PALTF.

**Figure 6 children-11-00691-f006:**
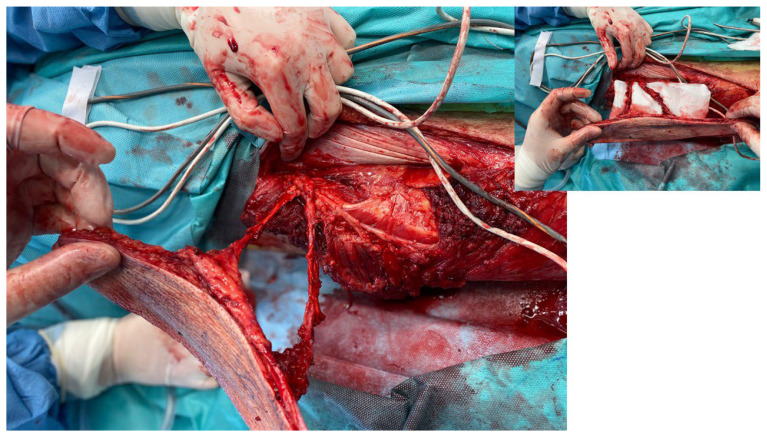
The patient with pedunculated musculocutaneous flap.

**Figure 7 children-11-00691-f007:**
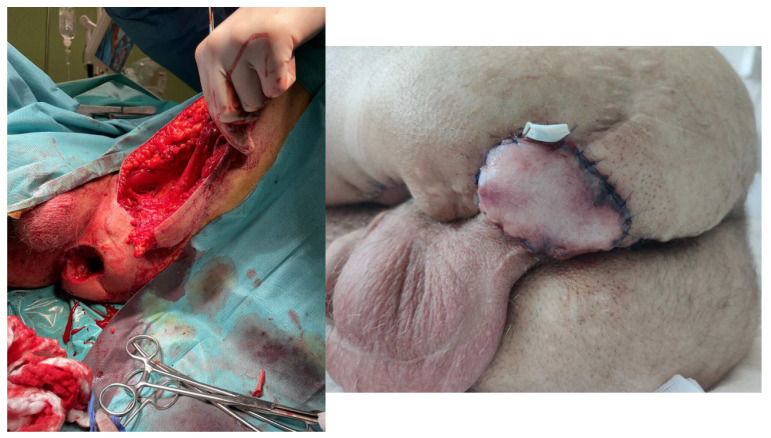
The patient with pedunculated musculocutaneous gracilis flap.

**Figure 8 children-11-00691-f008:**
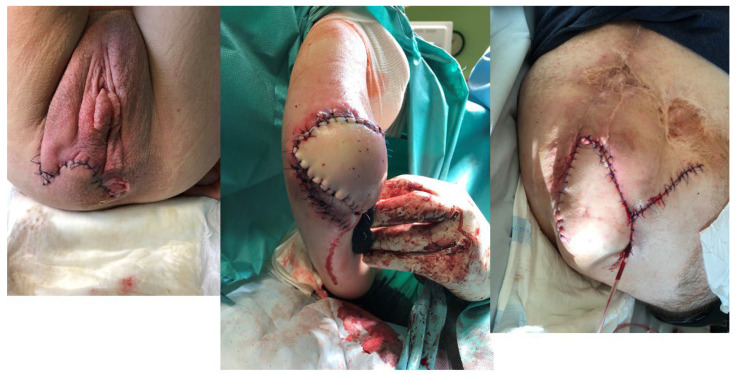
Examples of the locoregional and propeller flaps.

**Figure 9 children-11-00691-f009:**
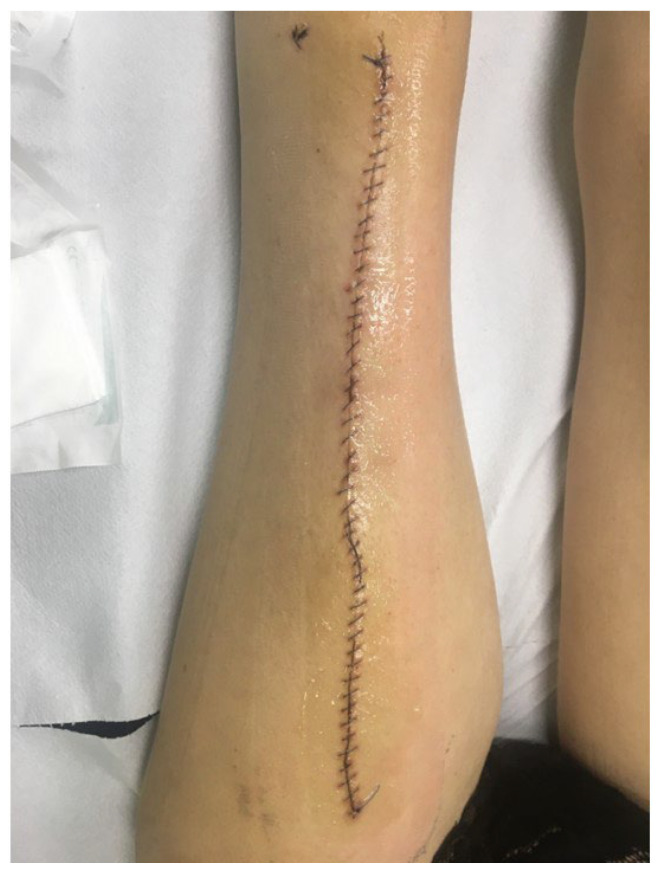
Donor site effect after pALTF flap.

**Figure 10 children-11-00691-f010:**
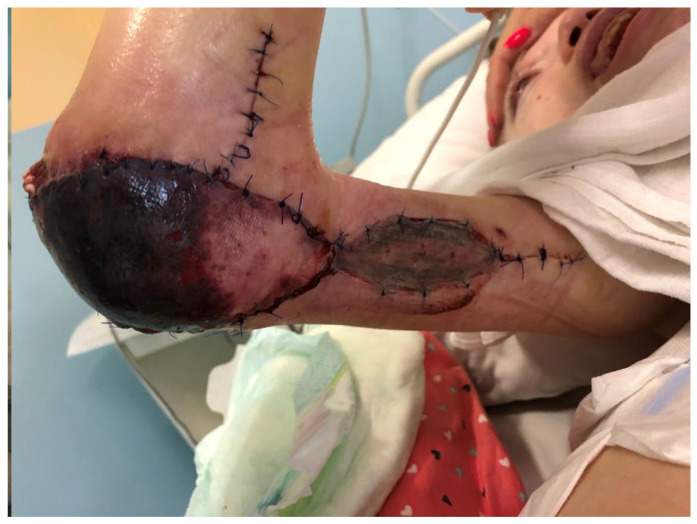
Partial necrosis of our propeller flap.

**Figure 11 children-11-00691-f011:**
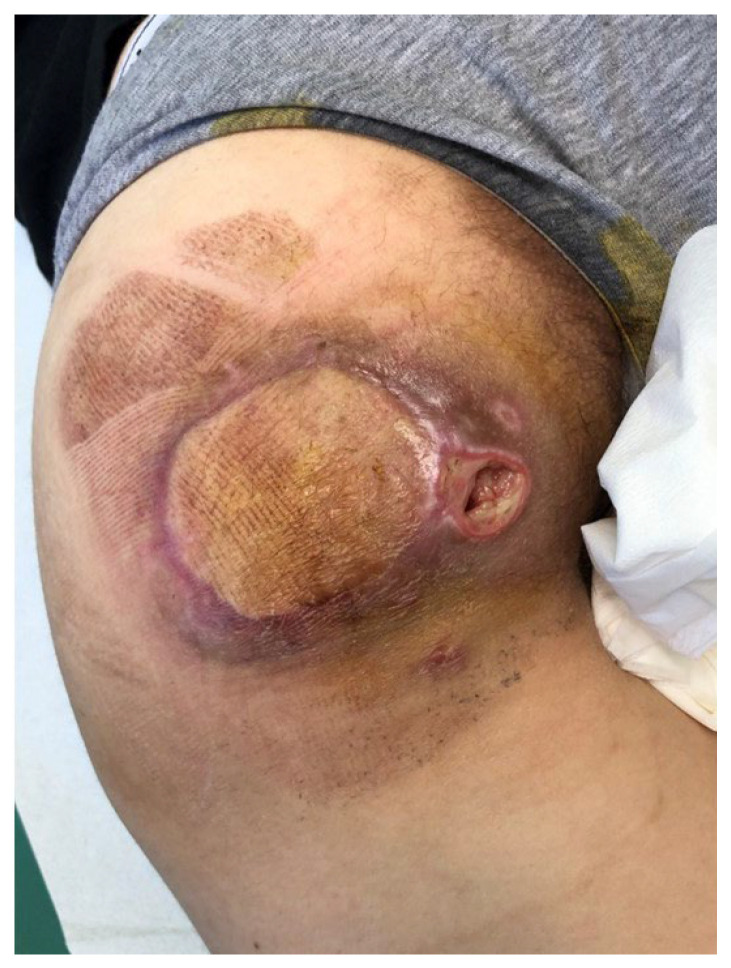
The only recurrence of PU in ischial region—after pALTF reconstruction.

**Figure 12 children-11-00691-f012:**
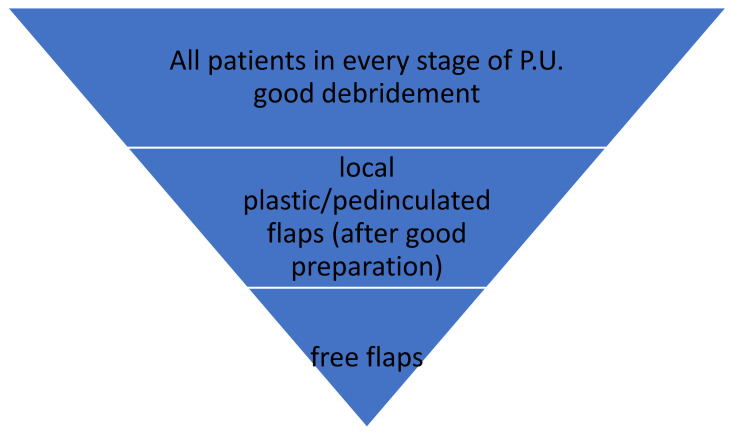
Reconstructive surgical ladder.

**Table 1 children-11-00691-t001:** Characteristics of patients, results of treatment. pALTF—Pedunculated anterolateral flap. pGracilisF—Pedunculated gracilis flap.

No. of Patient	Sex, Age	Site	Comorbidities/Diagnosis	Bedridden Status	Stage of PU	Time from Onset	Surgical Treatment	Recurrence and Other Complications	Earlier Surgeries
1	M, 14	ischial	MMC	yes	3rd	4 years	pALTF	NO	
2	F, 16	ischial	spinal injury	yes	4th	10 months	pALTF	YES, after 11 months—another surgery—ok	
3	F, 9	elbow	cerebral palsy	yes	3rd	1 year	local perforated flap	No, partial necrosis of flap—after debridement skin graft	
4	M, 16	stump of leg	amputation due to necrosis as a complication of cardiac disease	no	4th	1 year	propeller local flap	NO, long treatment with secretion of some fluid	
5	M, 15	ischial	MMC	yes	4th	3 years	pALTF	NO	
6	F, 17	sacral	MMC	no	3rd	11 months	local flap	NO, partial dehiscence of the wound	
7	M, 17	sacral	MMC	yes	4th	4 years	local plastic	NO	
8	M, 10	sacral	MMC	no	4th	2 years	Romberg plastic	NO	3 surgeries
9	F, 15	perineal reg	MMC	no	3rd	11 months	local plastics	NO, partial dehiscence of the wound, treated with hyperbaric therapy	
10	M, 15	ischial	MMC	partial yes	4th	2 years	pGracilisF	NO, partial dehiscence of the wound, treated with hyperbaric therapy	12 surgeries
11	M, 15	ischial	spinal injury diplegic	yes	4th	12 years	pGracilisF	NO, partial dehiscence of the wound, treated with hyperbaric therapy	

## Data Availability

Data are contained within the article.

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
