# Peer review of "Reconstructive Options for Pressure Ulcers in Pediatric Patients"

_children, 2024, doi:10.3390/children11060691_

Round 1

Reviewer 1 Report

Comments and Suggestions for Authors

This study on pressure ulcers is a very remarkable case series. However, I have a few suggestions to make the article better.

1.For the introduction, why don't you focus on this treatment and write details for the treatment. For example, it would be appropriate to share information about how the patient should be followed up after the operation and the sensitivity of the debridement.

2.I observed that you do not have a research question. Why don't you prepare a research question?

3. I saw that you gave information about the cases in the method section. I think it would be appropriate to prepare a table showing the demographic data of the cases (age, diagnosis, bedridden status, ties with the family, whether they have maternal care or receive special care). You can also present laboratory findings in a table. You can also provide a work flow chart. It will summarize your work. The journal also requests a summary form for the study.

Author Response

1. For the introduction, why don't you focus on this treatment and write details for the treatment. For example, it would be appropriate to share information about how the patient should be followed up after the operation and the sensitivity of the debridement.

  Reply: Thank You very much for this suggestion - I added more details for the treatment.

2. I observed that you do not have a research question. Why don't you prepare a research question?
  Reply: I added research question as well.

3. I saw that you gave information about the cases in the method section. I think it would be appropriate to prepare a table showing the demographic data of the cases (age, diagnosis, bedridden status, ties with the family, whether they have maternal care or receive special care). You can also present laboratory findings in a table. You can also provide a work flow chart. It will summarize your work. The journal also requests a summary form for the study.
  Reply:  I added a table with all necessary content - thank You so much for this point of view.

Reviewer 2 Report

Comments and Suggestions for Authors

The authors presented a small cohort of severe pressure sore conditions managed surgically, mainly with flaps.

Admittedly, this is a challenging condition. The experience among pediatric surgical centres is limited. The outcome presented in this manuscript is reasonable. 

Because of sample size and varied technique, no meaningful statical analysis could be made. 

To engage the interest of pediatric surgeons, maybe some key points of surgical techniques or post operative care can be discussed. Comparison to adult flap outcomes in similar situations may also be helpful. 

A flow char to algorithm in choosing your patients may also help the readers. 

Comments on the Quality of English Language

reasonable English

Author Response

The authors presented a small cohort of severe pressure sore conditions managed surgically, mainly with flaps.

Admittedly, this is a challenging condition. The experience among pediatric surgical centres is limited. The outcome presented in this manuscript is reasonable. 

Because of sample size and varied technique, no meaningful statical analysis could be made. 

To engage the interest of pediatric surgeons, maybe some key points of surgical techniques or post operative care can be discussed. Comparison to adult flap outcomes in similar situations may also be helpful. 

A flow char to algorithm in choosing your patients may also help the readers. 

Authors' Reply:

Thank You so much for your precise analysis.
I added some key points of surgical techniques in discussion I need to compare them th adults because there is no comparable group in pediatric population.
Thank You very much for this kind review.

Round 2

Reviewer 1 Report

Comments and Suggestions for Authors

The author seems to have made the necessary changes.